# Microfluidic-Based Technologies for Crossing the Blood–Brain Barrier Against Alzheimer’s Disease: Novel Strategies and Challenges

**DOI:** 10.3390/ijms26199478

**Published:** 2025-09-27

**Authors:** Irene Ferrari, Emanuele Limiti, Sara Maria Giannitelli, Marcella Trombetta, Alberto Rainer, Marcello D’Amelio, Livia La Barbera, Manuele Gori

**Affiliations:** 1Molecular Neuroscience Unit, Università Campus Bio-Medico di Roma, 00128 Rome, Italy; irene.ferrari@unicampus.it (I.F.); m.damelio@unicampus.it (M.D.); 2Advanced Technologies for Innovative Materials and Organ Models, Università Campus Bio-Medico di Roma, 00128 Rome, Italy; emanuele.limiti@unicampus.it (E.L.); s.giannitelli@unicampus.it (S.M.G.); m.trombetta@unicampus.it (M.T.); 3Microsystems & Nanotechnology Unit, Department of Engineering, Università Campus Bio-Medico di Roma, 00128 Rome, Italy; a.rainer@unicampus.it; 4Fondazione Policlinico Campus Bio-Medico di Roma, 00128 Rome, Italy; 5Department of Experimental Neurosciences, Istituto di Ricovero e Cura a Carattere Scientifico (IRCCS), Santa Lucia Foundation, 00143 Rome, Italy; 6Institute of Biochemistry and Cell Biology (IBBC)—National Research Council (CNR), International Campus “A. Buzzati-Traverso”, Monterotondo, 00015 Rome, Italy

**Keywords:** Alzheimer’s disease, blood-brain barrier, microfluidics, nanoparticles, organs-on-chips

## Abstract

Alzheimer’s disease (AD) represents the major cause of dementia worldwide, involving different etiopathogenetic mechanisms, but with no definitive cure. The efficacy of new AD drugs is limited by the multifactorial disease nature that involves several targets, but also by the difficult penetration across the blood–brain barrier (BBB) for reaching the target area at therapeutic doses. Thus, the inability of many compounds to efficiently bypass the BBB makes it arduous to treat the disease. Furthermore, the lack of more representative BBB in vitro models than conventional 2D cultures, and xenogeneic animal models that recapitulate AD pathogenesis, makes it even more difficult to develop definitive cures. In this context, microfluidics has emerged as a promising tool, offering advanced strategies for simulating the BBB, investigating its crossing mechanisms, and developing nanocarriers that successfully pass the BBB for brain-targeting, with particular interest in pathological states. The advantages of microfluidic platforms for studying the BBB role in pathophysiological conditions might herald more tailored and effective approaches based on functionalized nanosystems for treating AD. Here, we provide an overview of the latest advances in microfluidic-based technologies both for the synthesis of nanodrug delivery systems, and for developing advanced models of the BBB-on-a-chip to simulate this biological barrier, facing open challenges in AD, and improving our understanding of the disease.

## 1. Introduction

Alzheimer’s disease (AD) is the most common neurodegenerative disorder and the leading cause of dementia, accounting for 60–80% of cases among the elderly, particularly in developed countries [1]. As the global population ages and life expectancy rises, the number of individuals living with dementia in the world is currently estimated at over 50 million and is projected to nearly triple by 2050, as reported by the World Health Organization 2021. Despite its high prevalence and widespread impact, the exact causes of AD remain largely unknown, making this disease complex and challenging to treat.

AD is a multifactorial disease, and the risk of developing it is determined by a combination of factors, including environment, aging, and genetic predisposition. Pathologically, AD is characterized by the build-up of extracellular β-amyloid (Aβ) plaques and intracellular neurofibrillary tangles (NFTs), composed of hyperphosphorylated tau protein [2,3]. The improper clearance of these aggregates leads to neuronal toxicity, resulting in synaptic dysfunction, inflammation, and oxidative stress. This neuroinflammation further accelerates the disease progression and worsens cognitive symptoms. Over time, there is a progressive loss of neurons, particularly in limbic structures, which results in brain atrophy [4,5,6]. Despite significant research into AD pathology and potential interventions, many therapeutic strategies targeting Aβ plaques and tau tangles have not yielded satisfactory clinical outcomes in human trials, highlighting the disease’s intricate nature. Understanding the exact etiology of AD remains a significant challenge for developing effective diagnostic tools and treatments.

Numerous studies indicate that neurovascular dysfunction plays a key role in the onset and progression of AD, underscoring a strong association between cerebrovascular alterations and neurodegeneration [7]. Neuropathological alterations in blood vessels, which are common in the elderly, are strongly correlated with the classical hallmarks of AD, including NFTs and Aβ plaques [8]. Notably, over 90% of AD patients exhibit Aβ deposition in the small cerebral arteries and capillaries, a condition known as cerebral amyloid angiopathy (CAA) [9,10]. In addition, growing evidence indicates that blood–brain barrier (BBB) dysfunction and deterioration, closely associated with vascular risk factors, are among the pathophysiological features of AD [11].

The BBB is a vital interface that protects the brain by tightly regulating the exchange of substances between the circulatory system and the central nervous system (CNS). It acts as a shield, preventing toxins and pathogens from reaching the brain while allowing essential nutrients to pass through [12]. The fundamental structural and functional element of the BBB is the neurovascular unit (NVU), composed of capillary endothelial cells, astrocytes, pericytes, and neurons [13,14]. At the core of the physical barrier are the endothelial cells, arranged in a monolayer whose integrity is preserved by tight junction (TJ) and adherens junction (AJ) complexes [15]. Located within the capillary basement membrane, pericytes are essential for regulating cerebral blood flow, maintaining BBB stability, and guiding vascular development and angiogenesis. In addition, they participate in neuroinflammatory responses and exhibit stem cell-like properties [16,17]. Astrocytes signal to blood vessels to modulate blood flow according to the brain’s metabolic requirements through their endfeet, which closely contact capillaries. These glial cells also interact with pericytes to modulate BBB permeability under various physiological and pathological conditions [18]. As the brain’s primary cells, neurons communicate with astrocytes and interneurons, influencing signals to capillaries to adjust blood flow based on metabolic demands. Together, the components of the NVU orchestrate critical cell–matrix interactions and maintain overall tissue homeostasis, playing integral roles in processes such as angiogenesis and neurogenesis [19].

In AD, structural changes to the BBB, affecting endothelial cells, pericytes, astrocytes, and TJ, lead to increased BBB permeability [11]. A leaky BBB allows harmful substances, such as plasma proteins, cytokines, complement components, and immune cells, to enter the brain. These substances then activate inflammatory pathways and increase the activity of γ-secretase, creating a self-amplifying cycle of neuroinflammation and Aβ deposition (Figure 1) [20,21]. Additionally, BBB disruption impairs Aβ transport by primarily reducing efflux mediated by Low-Density Lipoprotein Receptor related Protein 1 (LRP-1) [22], and enhancing influx mediated by Receptor for Advanced Glycation End-products (RAGEs), thereby further promoting Aβ accumulation [23,24]. All these factors create a harmful feedback loop linking BBB dysfunction, Aβ pathology, and cognitive decline.

One of the major challenges in developing effective therapeutic strategies lies in the limited ability of drugs to reach the brain in therapeutically relevant concentrations, due to the highly selective nature of the BBB. This protective function, therefore, represents a significant obstacle to delivering drugs to the CNS, as it effectively excludes ~100% of large-molecule neurotherapeutics and over 98% of small-molecule drugs. Only lipophilic molecules with low molecular weight can readily cross this barrier, significantly restricting the treatment options for most CNS disorders [25,26,27]. This limitation is a key factor in the discrepancy between promising preclinical research and less successful clinical outcomes in the treatment of AD. It highlights the need for more accurate preclinical models and a better understanding of the BBB complexity to develop strategies to enhance drug delivery. Conventional in vitro BBB models, such as two-dimensional (2D) cultures or Transwell inserts, and xenogeneic animal models, fail to fully recapitulate the complexity and pathophysiology of the human BBB, thereby limiting translational progress [28]. Dynamic flow, multicellular architecture, and human-specific transport kinetics are necessary to accurately predict drug behavior in vivo.

Emerging microfluidic technologies offer promising solutions to these challenges. These platforms enable the development of dynamic and physiologically relevant BBB-on-chip models, which can be used to investigate transport mechanisms, simulate AD-specific conditions, and evaluate therapeutic delivery in a more predictive manner [29,30]. By integrating endothelial cells, pericytes, and astrocytes under controlled force stress, these devices recreate key aspects of the NVU, enabling real-time study of Aβ transport, and inflammatory responses in AD [31]. Simultaneously, microfluidic systems support the controlled synthesis of functionalized nanocarriers, designed to enhance drug permeability across the BBB, to achieve targeted delivery to affected brain regions, thus improving drug release kinetics [32].

In this review, we first examine the benefits of BBB-on-a-chip devices to mimic both healthy and diseased barrier properties, focusing on their utility for modeling the BBB pathological state in the framework of AD. Then, we survey recent advances in microfluidic-assisted design of functionalized nanoparticles, highlighting how material choice and surface modifications influence BBB crossing and cellular uptake, evaluating nanocarrier drug transport and therapeutic efficacy in the CNS. Finally, we discuss emerging strategies to improve cell-specific targeting through nanoparticle-mediated drug delivery, and to harness microfluidic-based approaches into clinical settings for novel AD treatments.

## 2. Models for Studying the BBB

The development of BBB models has been driven by the need to explore the brain microenvironment and develop new therapies for neurodegenerative diseases. In vivo models have been instrumental in studying the BBB and its involvement in neurological disorders, drug delivery, and therapeutic strategies. These models, specifically rodents, reflect the complex biology of a living organism and offer several advantages, including the generation of robust and physiologically relevant data, the preclinical assessment of novel therapeutics, the simultaneous investigation of immune responses to new compounds, as well as drug pharmacokinetics and pharmacodynamics [33]. However, there are some differences between species in BBB composition, including variations in TJ proteins, transporters, and immune system interactions that can limit a direct translation to human models. In this context, xenogeneic animal models are valuable in bridging the gap between experimental models and clinical applications, as they enable the investigation of BBB behavior in the context of human tissues within a living organism. Nevertheless, these models have significant limitations. Indeed, they require immunodeficient animals, which means that the interaction between human tissues and the immune system is unphysiological. Furthermore, differences in cell junctions, transporters, and cytokines between species can affect the predictability of results [34,35]. Lastly, setting up xenogeneic models is costly, technically complex, and subject to variability in factors such as the type of implant, cell line, and host condition. All these factors limit reproducibility and throughput and raise important ethical and regulatory issues.

### 2.1. Current In Vitro Models of the BBB: Advantages and Limitations

Each BBB model is basically devised and manufactured to facilitate the exploration of the brain microenvironment and the development of new therapeutic treatments for neurodegenerative disorders. Currently, there is no perfect or ideal in vitro BBB model, primarily due to the difficulty of simulating all in vivo physiological conditions [36]. Selecting the most appropriate method is therefore critical and depends on the experimental requirements and the desired clarity of the results.

Research in this field began in 1953 with the introduction of the Transwell systems, constituted by a monolayer of cells cultured on a porous filter membrane [37]. After the first approaches using endothelial embryonic cells, the isolation of primary cerebral microvessels enabled the incorporation of brain-specific endothelial cells into these models [38,39], thereby improving their physiological relevance. This simple model was used for a variety of studies, particularly the analysis of drug permeability. Indeed, the Transwell system allows for easy and reproducible assessment of TJ formation by measuring trans-endothelial electrical resistance (TEER) between the two chambers, while simultaneously monitoring permeability to tracer molecules, such as fluorescent dyes, radiolabeled sucrose, or dextran [40]. In addition, this system is extremely accessible and versatile due to the commercial availability of inserts with calibrated porosity and standardized protocols. However, despite its simplicity, this basic monoculture model had several limitations. Indeed, it poorly replicated the in vivo BBB environment, as endothelial cells cultured on the apical membrane exhibited minimal BBB-specific characteristics. Notably, the absence of well-formed TJs resulted in abnormally high permeability to molecules like sucrose [41]. Consequently, the use of this model for BBB-related studies yielded poor reliable results, which could not be translated into physiological conditions. In the 1980s, growing knowledge of TJ protein expression, particularly in endothelial cells and astrocytes, led to the development of co-culture Transwell systems. Indeed, the co-culture of endothelial cells and astrocytes improved the TJ organization and increased the BBB structural and functional resistance [42,43], making it a more reliable system. Similarly, co-culture with pericytes promoted endothelial cell maturation and contributed to the formation of a tighter, less permeable barrier [44]. However, some studies have reported that the presence of pericytes could lead to BBB disruption and increased permeability, highlighting the complexity of this system [45,46]. Thus, these models were later refined with the addition of both astrocytes and pericytes, resulting in more complex and physiological triple co-culture systems [47,48]. However, while these models have provided valuable insights into the structure and function of the BBB, none of them fully replicated the complexity of the in vivo condition. For example, the static nature of the cells means that there is no laminar or pulsatile shear stress, which is characteristic of cerebral capillaries. Shear stress is a crucial physiological stimulus that actively drives the differentiation of BBB endothelial cells. In its absence, endothelial cells fail to fully differentiate and do not acquire a physiological phenotype [49]. Dynamic in vitro BBB models using human brain microvascular endothelial cells (HBMECs) show that shear stress significantly enhances the RNA expression of key TJ and AJ components, including zonula occludens-1 (ZO-1), claudins 3 and 5, cadherins, catenin α2 and β1, and actin α2 [50]. Importantly, actin and catenin directly interact with transmembrane proteins to assemble TJ and AJ complexes, thereby strengthening inter-endothelial connections and enhancing barrier selectivity. Under shear stress, HBMECs exhibit greater maturation and differentiation, as evidenced by an increase in TEER from 100 to 700 Ω·cm^2^. Shear stress also induces the expression of endothelial drug transporters and metabolic pathways that enhance the BBB’s ability to protect the CNS from harmful substances [13,50]. Furthermore, laminar shear stress elicits vascular protective responses in human cerebral microvascular endothelial cells (hCMEC/D3), including activation of the NRF2 transcription factor, which elevates intracellular glutathione levels and protects cells from oxidative stress. Additional benefits include the inhibition of pro-inflammatory responses, cell motility, and proliferation [51]. Together, these effects highlight the important protective role of shear stress in maintaining BBB cells quiescent and functionally intact. Another limitation is the 2D geometry, which is inadequate in reproducing the cylindrical lumen and the correct BBB volume ratio. The membrane-to-plastic interface has been identified as the location of an “edge effect”, which is defined as a phenomenon that fosters non-physiological paracellular leaks. This edge effect is known to artificially lower measured TEER values, typically to only a few hundred Ω·cm^2^ (100–500 Ω·cm^2^), well below the in vivo measurements [52,53]. Due to these constraints, Transwell systems remain ideal only for preliminary permeability screenings and targeted molecular studies (Table 1). Therefore, the creation of physiologically relevant in vitro platforms that can accurately mimic BBB phenotypes and enable real-time monitoring of barrier integrity and neurodegenerative processes is critical to advancing our understanding of neurological disorders and for developing targeted therapies.

Technological advancements from the late 2000s onwards have facilitated the development of dynamic, three-dimensional (3D) models, such as cerebral organoids and microfluidic organ-on-a-chip (OoC) systems, which permit the introduction of flow and shear stress and produce more realistic vascular geometries [54]. The BBB organoid models derive from embryonic stem cells (ESCs), pluripotent or induced pluripotent stem cells (iPSCs), or primary brain cells, which differentiate into the main cellular components of the BBB. These components self-organize into 3D structures, accurately mimicking the cellular architecture and microenvironment of the human NVU [55,56,57,58]. These models are typically constituted of brain microvascular endothelial cells, astrocytes, and pericytes, and in some cases, neurons and microglia, enabling the study of dynamic cell–cell interactions. Compared to the 2D in vitro systems described above, the BBB organoids exhibit more robust TJ formation, enhanced barrier integrity, and more physiologically relevant expression of transporters and receptors [59]. In addition, one of the major advantages of the BBB organoids is their ability to reflect human-specific and even patient-specific genetic and epigenetic backgrounds, so that patient cells can also be used to produce BBB organoids. This enables personalized studies of barrier dysfunction in neurodegenerative or neuroinflammatory diseases [60]. Nevertheless, organoid models encounter significant obstacles (Table 1). Consistent size and cellular composition are technically challenging to achieve. The absence of a microvasculature can severely compromise cell survival, tissue development, and physiological relevance as organoids grow. Without a vascular network, the delivery of oxygen and nutrients is restricted, resulting in hypoxic stress and the upregulation of hypoxia-inducible markers, such as HIF-1. Over time, the limited diffusion of oxygen and nutrients into the core of the organoid results in necrosis during prolonged culture [61]. In addition, the absence of true perfusion means that endothelial cells are deprived of essential shear stress. Consequently, organoids exhibit incomplete maturation of BBB endothelial cells and defects in TJ and AJ complexes. These defects compromise BBB integrity and restrict the ability of the organoids to replicate the physiological phenotype and function of the barrier [62,63,64]. Therefore, they usually form immature, non-perfused microvessels that rarely develop permeable lumens or achieve the high TEER values observed in vivo. Furthermore, quantitative permeability assays are hindered by the solid architecture of the spheroids, which makes accessing the inner core both invasive and imprecise. High variability in organoid size, cell composition, and vascularization between batches further compromises reproducibility and standardization [65]. Finally, the extended timeframes required for cell differentiation (ranging from weeks to months), combined with the high cost of specific cell culture media, growth factors, and equipment, significantly limit their throughput compared with more efficient alternatives such as microfluidic platforms.

The microfluidic OoC systems, on the other hand, allow scientists to harness more physiologically accurate representations of the BBB, also for long-term cultures, by incorporating, among others, vascular flow and shear stress.

Current organ and tissue culture models increasingly focus on vasculogenesis and the optimization of microfluidic architecture to replicate the structural and functional characteristics of the human BBB more accurately.

**Table 1 ijms-26-09478-t001:** Summary of the advantages and limitations of in vitro BBB models.

BBB Model	Advantages	Limitations
Transwell systems [40,41,52,53]	- Quick setup for permeability assays - TJ formation can be easily and reliably assessed by measuring TEER - Cost-effective and easy to use - High-throughput screening compatibility	- Poor mimicry of in vivo endothelial conditions - The system is static and lacks physiological shear stress - Limited cell differentiation and polarization - Abnormally high permeability measurements
Co-Culture Models [47,48,49]	- Better mimics cell-to-cell interactions - A more biologically relevant environment - Low cost	- Static culture system and absence of physiological shear stress - Limited cell differentiation and polarization - Inadequate reproduction of cylindrical lumen - High permeability measurements
3D Culture Models (Organoids) [55,56,57,58,59,60,61,62,63,64,65]	- Accurate mimicry of the cellular architecture and microenvironment of the NVU - Study of dynamic cell–cell interactions - More physiologically TJ formation and barrier integrity - Personalized studies of barrier dysfunction	- Poor perfusion and nutrient delivery - Limited control over microenvironment - Absence of physiological shear stress - High variability in size, cell composition, and vascularization between batches - High costs and long timescales

### 2.2. Microfluidic Models of the BBB in the Setting of AD

As previously detailed in Section 2, the gold standard to investigate neurodegenerative disorders associated with the BBB is based on in vivo animal models or in vitro cellular ones [12,66]. Animal models are usually preferred because of their better representation of the complex physiological and pathological behavior of human BBB. However, these models suffer from several limitations that may impede their use in preclinical studies. To begin with, the ethical issues associated with the use of animal models in scientific research, in accordance with the 3R principles (i.e., Replace, Reduce, Refine), may represent an obstacle that demands for the development of alternative in vitro models [67,68,69]. Furthermore, the high cost, low throughput, and inefficiency resulting from inter-species biological differences necessitate efforts to develop novel strategies to recreate relevant biological models of the BBB [70,71]. On the other hand, current in vitro models are characterized by their robustness, reproducibility, ease of use, and the capacity for rapid and extensive analysis. However, these models are inadequate in replicating the complex physiology of the BBB on a 3D scale (Table 1), thereby hindering their experimental relevance [72].

Over the past decades, microfluidics has emerged as a promising field of research for investigating the fluid dynamics of nano- to picoliter-scale fluid volumes inside microchannel devices [73]. Recent advances in this area have enabled the development of micro-engineered platforms designed to better replicate the 3D architecture and function of human organs and tissues in vitro, known as OoC. When applied to modeling the BBB, these OoC offer enhanced physiological relevance and greater predictive accuracy compared to traditional Transwell-based models. Notably, the integration of shear flow dynamics within the vascular compartment, combined with the geometrical adaptability of these devices, enhances their capacity to reach an unprecedented level of biomimicry in BBB in vitro modeling [74,75,76].

In general, the BBB-on-a-chip technology is based on three main structural components: (i) the vascular chamber, designed for the cultivation of the endothelial cell layer (e.g., microvascular endothelial cells) and the replication of blood flow through continuous perfusion, (ii) the cerebral chamber, where neurons, astrocytes, microglia and pericytes are cultured to mimic the brain microenvironment, and (iii) a separating interface between the two chambers, which may consist of a porous membrane or a microstructured array of channels or micropillars, allowing controlled communication between the compartments.

Based on the structural organization of its core components, three different BBB-on-a-chip configurations can be identified: (i) vertical design, (ii) parallel design, and (iii) full-contact design (Figure 2). The key distinguishing feature among these configurations lies in the positioning of the separating interface relative to the vascular and cerebral chambers [31,77].

In the vertical design (Figure 2(Ai)), a porous membrane separates two stacked chambers, offering a high contact surface. Typically, the upper chamber houses the endothelial cells representing the vascular compartment, while the lower chamber is deputed to mimic the neural parenchyma. In contrast, the parallel design (Figure 2(Aii)) positions the vascular and cerebral chambers side-by-side in a horizontal configuration, replacing the porous membrane with microchannels or micropillars. This design has lower contact area compared to the vertical one, but the reduced vertical projection improves optical accessibility, facilitating the high-resolution imaging of the cultured cells.

The full-contact design (Figure 2(Aiii)) represents the most promising approach with higher level of fidelity to the actual BBB structural organization. Here, circular lumen embedded into a 3D hydrogel within microfluidic channels enables the endothelial cell culture in a tubular structure that closely mimics the actual in vivo arrangement. Moreover, this configuration is used to maximize the contact area and better reflect the actual fluid dynamics of the BBB.

Recently, growing efforts have been directed toward leveraging the advantages of BBB-on-a-chip models. As an example, the work led by Tanzi and coworkers has significantly advanced the development of microfluidic models to better understand the role of the BBB in the context of AD. In a first attempt [78], researchers developed a 3D triculture system combining human neurons, astrocytes, and adult microglia within a two-chamber cylindrical microfluidic device in a parallel configuration (Figure 2B). The central chamber hosted human neural progenitor cells (ReN cell) embedded in Matrigel, while microglia were seeded into angular side chambers. Microchannels connecting the chambers allowed microglia cells to migrate toward the central channel in response to chemokines and cytokines expressed by neurons and astrocytes, such as CCL2, IL8, TNF-α, and IFN-γ. The proposed device enabled the study of microglia recruitment, neuroinflammation, Aβ accumulation, and p-tau aggregation mimicking an AD model. Although this model represents some of the key features of AD pathogenesis, it lacks a BBB compartment, therefore limiting its biological relevance. Successively, a major step forward was taken with the integration of a BBB-like structure into a five-channel parallel microfluidic platform (Figure 2C) [79]. The device featured a 3D neuronal culture derived from the same ReN progenitor cells on one side, and a tubular monolayer of hCMEC/D3 brain endothelial cells on the other, separated by a collagen-filled channel acting as an interface. This setup allowed for controlled cross-talk between neural and vascular compartments. The endothelial barrier exhibited key BBB characteristics—such as TJs (claudin-5, ZO-1), adherens junctions (VE-cadherin), and Aβ transporters (RAGE, P-gp, LRP1)—and showed increased permeability and Aβ deposition under AD conditions. The modular culture protocol also enabled sequential maturation of neural and vascular components.

Recently, researchers succeeded in developing a self-assembling, perfusable 3D neurovascular system designed within a multi-compartment microfluidic chip in full-contact configuration (Figure 2D) [80]. Here, primary human brain endothelial cells, pericytes, and astrocytes were embedded in a fibrin gel to form vascular networks, which developed around a central neural compartment containing ReN cells, expressing familial AD mutations, differentiated into neurons after seeding. The design included side perfusion channels to introduce the flow component, mimicking both interstitial and vascular dynamics. This setup supported long-term co-culture and enabled high-resolution tracking of AD-related changes in BBB function, such as progressive increase in permeability, altered endothelial marker expression, vascular structural changes, and Aβ accumulation in both neuronal and vascular compartments. Importantly, this neurovascular system-on-a-chip allows for compartment-specific sampling, offering valuable insights into therapeutic screening and mechanistic studies.

Additionally, BBB-on-a-chip models offer novel platforms with increased fidelity to study a series of AD-related phenomena. In particular, it has been demonstrated that dysregulation of the calcineurin–nuclear factor of activated T cells (NFAT) signaling pathway, in combination with APOE alterations in pericyte-like mural cells, promotes APOE4-associated CAA. The obtained results highlighted the key role for pericytes in the pathogenesis of APOE4-driven CAA and the calcineurin–NFAT pathway as a potential therapeutic target in CAA and AD [81]. Moreover, a neurovasculature-on-a-chip model was developed to investigate how diabetes mellitus may contribute to AD. The model showed that hyperglycemia disrupts BBB integrity, promotes Aβ accumulation, and increases Tau phosphorylation. SIRT1 emerged as a key regulator of these processes and could be modulated through genetic and pharmacological approaches [82]. Another BBB-on-a-chip model was used to study immune-brain interaction in AD. In this work, the increased infiltration of AD-derived monocytes compared to healthy controls was demonstrated to promote neuroinflammation and neuronal apoptosis. Treatment with Minocycline and Bindarit agents reduced monocyte infiltration and attenuated both astrocyte activation and neuronal death [83].

Table 2 recapitulates the technological setup and application scenarios of the presented BBB-on-a-chip models. As previously discussed, these devices represent a significant advancement toward the establishment of biorelevant OoC models. These models are able to mimic not only the structural organization of the intended tissues or organs, but also to provide deep insights into specific biological features of AD. Therefore, these platforms represent an outstanding achievement in this field, establishing the foundation for their extensive use in preclinical disease modeling and drug testing.

### 2.3. Regulatory and Ethical Issues of OoC

In the context of regulatory oversight, OoC translation into the clinical setting still needs a thorough process of standardization for their safety and efficacy compared to the current state-of-the-art in vitro and in vivo preclinical models, in particular for the healthcare and pharmaceutical fields. Taking into account the high level of complexity of the nervous system and brain physiology, faithfully replicating the BBB and brain structure and function into a chip represents a great challenge, requiring a rigorous validation process versus current 2D and 3D cell culture models, which necessarily involves functional optimization, consistency, and reproducibility of research results. The first significant attempts in this direction were represented by two remarkable works performed by Emulate Inc. for predicting species-specific toxicities of hepatotoxic compounds, and for a toxicology screening that compared outcomes from hepatic organoids and mouse models with liver-on-chips, to better evaluate risk assessment and human relevance of microfluidic liver models [84,85]. Besides aligning and matching the outcomes of specific OoC models among different laboratories worldwide (i.e., reproducibility), the other requirement for streamlining and accelerating preclinical drug testing using this emerging technology is the result consistency between animal models and OoC models, with the final goal to reduce and possibly replace the former. As a matter of fact, one of the most significant advantages offered by OoCs is their ethical impact together with a more precise control over neural activities in brain-related research, using animal- or human-derived cells. Thus, in the near future, as a valid alternative to animal testing, these novel platforms should help curb expensive and time-consuming drug failures in the clinic for drug development, safety tests, and targeted therapies [86]. To this aim, OoC systems, including the BBB-on-a-chip, are required to meet rigorous safety standards set by regulatory agencies, such as the FDA and EMA [87]. In 2020, the US FDA became the first government agency to consider the OoC technology as a potential drug development tool for assessing the risk of hepatotoxicity of certain drug candidates in humans by launching a pilot program for the liver-chip validation into the regulatory applications for new medical products [88]. At the EU level, in 2022, the EU reference laboratory for alternatives to animal testing (EURL ECVAM) produced a catalog of resources for inventors and end-users of this new technology to promote, validate, and enable its development and use in the scientific community, which represents the first step towards OoC standardization. To address regulatory challenges, this catalog, developed and constantly updated in collaboration with the Regulatory Advisory Board (RAB) of the European Organ-on-Chip Society (EUROoCS), includes all the main regulatory fields of interest and OoC applications [89].

## 3. Microfluidic Synthesis of Nanocarriers for AD Therapy

Over the past few decades, the integration of nanotechnology into the biomedical sciences has gained substantial momentum, culminating in the emergence of nanomedicine as a distinct and rapidly advancing field of research [90,91].

In this context, nanoparticles (NPs) represent one of the most promising classes of materials for therapeutic, diagnostic, and theranostic applications. NPs are typically classified according to their composition into three main categories: (i) organic NPs (polymeric- or lipid-based NPs) [92,93,94], (ii) inorganic NPs [95,96,97], and (iii) hybrid NPs [98,99]. Despite their structural differences, most NPs share a set of desirable characteristics: they can encapsulate and deliver active compounds, are generally biocompatible, exhibit low toxicity, and can be easily functionalized. In addition, thanks to their small size and ability to bind specific ligands, NPs are capable of crossing biological barriers, such as the BBB, becoming favorable candidates for the treatment of CNS disorders [100,101].

The introduction of microfluidics represents a change in paradigm in the synthesis of NPs, addressing the limitations related to traditional batch processes—limited batch-to-batch reproducibility, low scalability, scarce control over process parameters, and high consumption of reagents. In microfluidic systems, NPs can be produced via two main approaches: single-phase flow synthesis and droplet-based synthesis [102,103,104].

In single-phase systems, NPs synthesis is commonly achieved through in-flow nanoprecipitation. In this process, precursors are first dissolved in a suitable solvent and then subjected to microfluidic mixing with an antisolvent to induce nanoparticle formation through three main stages: nucleation, growth, and precipitation. Unlike batch processes, where these steps occur simultaneously, resulting in uncontrolled growth and broader size distribution, microfluidics enables their spatial separation along the flow path, offering finer control over particle size and properties [105].

Conversely, droplet-based microfluidic processes rely on the generation and manipulation of emulsified systems (such as water-in-oil W/O or oil-in-water O/W emulsion). By confining reactions within microscale droplets or plugs, this technique significantly reduces reaction volumes, enhances mixing efficiency, and improves the reproducibility and uniformity of NP synthesis [103].

Precise control over the NP size and polydispersity afforded by microfluidics renders these systems particularly suitable for the production of nanomaterials capable of crossing the BBB, and potentially treating neurodegenerative diseases [106,107]. Despite several recent reviews that have broadly covered the use of microfluidic technologies to synthesize NPs that cross biological barriers [108,109,110,111], in this work, we focus specifically on microfluidically synthesized NPs designed for the treatment of AD. However, to the best of our knowledge, there are no studies that integrate microfluidically synthesized NPs into BBB-on-a-chip models specifically designed to replicate the pathological features of AD described in previous sections. While many studies have evaluated the permeability of the BBB by NPs in AD models [29,107,112,113,114,115], and in spite of the growing interest in both microfluidic synthesis of NPs and 3D OoC platforms, there seems to be a lack of convergence between these two promising biomedical areas in the context of AD research. This gap highlights an important opportunity for future studies to explore more physiologically relevant in vitro models that could better predict the therapeutic efficacy and brain permeability of NP-based systems under AD-like conditions.

A summary of the relevant studies is provided in Table 3.

The reviewed studies identify two key therapeutic targets as the most promising for NP-based treatment of AD: (i) extraneuronal aggregates of Aβ, and (ii) modulation of cellular and microenvironmental mechanisms of neurodegeneration, aiming to prevent neuroinflammation and promote neuroprotection.

In particular, the fibrillation of Aβ, with consequent aggregation and plaque deposition, can be effectively inhibited by the action of D1 peptide, as reported by N. Hassan et al. [116]. In this study, the authors developed peptide-functionalized magneto-plasmonic NPs (MFMPs), starting from positively charged chitosan-coated gold nanorods (AuNR-Chit), and negatively charged D1-functionalized iron-based magnetic NPs (MNP-D1). They investigated the microfluidic synthesis of these nanocomplexes, which yielded superior performances compared to conventional batch mixing in terms of particle size (lowering the DLS measured size from >900 nm for the bulk synthesis to <100 nm for the microfluidically templated nanocomplexes), polydispersity, and both plasmonic and magnetic properties. Moreover, the resulting NPs showed good biocompatibility in a wide range of concentrations (0.0035 to 0.058 nM) in an SH-SY5Y neuroblastoma cell model. Notably, a marked reduction in Aβ fibril formation (77.8%) was obtained when Aβ monomers were incubated with MFMP-D1. This inhibitory effect was attributed to the interaction between NPs and Aβ peptides, which prevents Aβ aggregation in vitro.

In a more recent study, Yanyan Xu et al. [117] tried to deal with the same biological target by co-delivering β-site APP cleaving enzyme 1 (BACE1) siRNA (siB) together with a-mangostin (α-M) in a synergistic manner, combining neuroprotection, microglia reprogramming, and Aβ inhibition. Here, the microfluidically synthesized lactoferrin (Lf)-functionalized lipid nanoparticles (LNPs) present multiple advantages: (i) they enable the co-encapsulation of liposoluble small-molecular therapeutics (α-M) and hydrophilic nucleic acid (siB) with high encapsulation efficiency (91.85% for siB and 35.64% for α-M, respectively), and (ii) the Lf functionalization can target the Lf receptor commonly overexpressed in AD neurons and in respiratory epithelial cells, making these devices suitable for intranasal delivery. The microfluidic process results in the robust formulation of 107.89 ± 1.34 nm NPs with a surface charge of −10.62 ± 5.48 mV. Flow cytometry analyses revealed rapid cellular uptake of Lf-LNPs in both PC-12 and Calu-3 cells within 4 h post-incubation. Moreover, to evaluate their potential to promote Aβ elimination by microglial cells, BV-2 cells, an intracerebral Aβ-scavenging microglial model, were treated with Lf-LNPs loaded system for 24 h. The results demonstrated a significantly augmented uptake of FITC-Aβ, attributed to α-M induced overexpression of low-density lipoprotein receptor (LDLR), a key mediator of Aβ clearance. In parallel, siB-loaded LNPs were evaluated for their efficiency in downregulating BACE1 expression. The system achieved a robust silencing effect, with a knockdown efficiency of 66.38%. These results confirmed the ability of these systems to knockdown BACE1 expression and to regulate Aβ elimination via microglial reprogramming. Finally, in vivo studies were conducted in APP/PS1 transgenic mice. Behavioral experiments, such as the Morris Water Maze (MWM) and Y-maze test, revealed marked cognitive improvement in mice treated with Lf-LNPs compared to untreated transgenic controls. This cognitive recovery correlated with a significant reduction in Aβ plaque burden, accompanied by decreased neuroinflammation and oxidative stress.

Neuroprotection and neuroinflammation were also addressed by Fariba Mohebichamkhorami et al. [118], who developed hybrid NPs composed of chitosan (CS) and graphene quantum dots (GQDs) synthesized via a microfluidic approach. Specifically, a CS/GQDs solution was injected into the central stream of a cross-flow microfluidic device, while the CS crosslinker, tripolyphosphate (TPP), was introduced through the lateral channel. The precise control offered by microfluidic synthesis allowed the formation of ultrasmall (DLS measured size range: 10–20 nm), uniformly distributed nanoparticles suitable for IN administration. In vitro assays confirmed the biocompatibility and cellular internalization of CS/GQDs NPs in the glioma C6 cell model, while in vivo experiments using a streptozotocin (STZ)-induced rat AD model demonstrated significant functional and histological improvements. In particular, behavioral testing using the Radial Arm Water Maze (RAWM) showed relative memory improvement in treated animals. Moreover, TEM analysis after NP treatment confirmed their cellular uptake and their localization in the hippocampus. However, NP treatment did not affect Aβ plaque deposition in the intercellular environment, nor did it affect MAP2 and NeuN expression, indicating no impact on neuronal cell proliferation and regeneration. Therefore, researchers attributed the observed positive impact on memory primarily to the NP anti-inflammatory and anti-oxidative stress properties, which prevent neuroinflammation and neuronal death.

In the study proposed by Yisheng Chen et al. [119], they microfluidically synthesized transferrin-functionalized liposomes loaded with pantothenate (Pan@TRF@Liposome NPs), with a size of about 50 nm and surface charge of −25 mV for the treatment of AD. In particular, the Authors aimed to tackle the nuclear translocation of pyruvate kinase M2 (PKM2) mediated by Chromosome Region Maintenance 1 (CRM1), which causes the upregulation of pro-inflammatory and pro-apoptotic genes in AD. The mechanism of action of pantothenate is known to reduce oxidative stress and mitigate the CRM1 activity, thus inhibiting the PKM2 nuclear translocation. Treatment of APP/PS1 transgenic mice with Pan@TRF@Liposome NPs, combined with physical exercise, resulted in enhanced cognitive performances (MWM and Y-maze test), noticeable reduction in the expression of markers of neuroinflammation (TNF-α, IL-1β, IL-6), oxidative stress, and neuronal apoptosis, and preservation of NeuN and MAP2 expression, indicating protection of neuronal structure and function. This study demonstrates that inhibition of PKM2 nuclear translocation by Pan@TRF@Liposome NPs offers a promising strategy to slow down the progression of AD, with Pantothenate that acts as a metabolic and anti-inflammatory modulator, effectively disrupting a central regulatory pathway in the neurodegenerative cascade.

It is worth noting that the microfluidic approach is not only limited to NP formulation, but has also been validated for the production of micron-sized hydrogel particles intragastrical administered for the treatment of AD [120,121,122].

Yao et al. [120] employed a microfluidic electrospray technique to fabricate alginate–dopamine core–shell microcapsules with strong adhesion to the intestinal mucosa and prolonged Donepezil release. In APP/PS1 transgenic mouse models, the treatment significantly reduced AChE activity, resulting in improved behavioral performance. Building on the same microfluidic technique, Zhou et al. [122] produced alginate–dopamine core–shell microcapsules encapsulating mesoporous silica nanocarriers loaded with galantamine. This hierarchical design allowed for controlled and pH-responsive release in the gastrointestinal tract, resulting in improved cognitive behavior, modulation of AChE activity, and reduced Aβ plaque deposition in vivo. Successively, Zhang et al. [121] used a droplet-based microfluidic device to synthesize PLGA–GelMA core–shell microcapsules loaded with Ginkgo biloba extract. These microparticles exhibited high monodispersity and encapsulation efficiency, and their oral administration led to reduced Aβ plaque deposition and marked cognitive improvement in AD mice.

For completeness, Table 3 provides an overview of several DDSs investigated in diverse CNS conditions other than Alzheimer, including glioblastoma and other brain tumors, epilepsy, post-traumatic brain injury, neuroinflammation, ischemic stroke, and neurodegenerative disorders [123,124,125,126,127,128,129,130,131]. Notably, most reported formulations rely on lipid-based [126,128,129,130] or polymer–lipid systems [123,124,125,131], with a particular tendency, highlighted in more recent approaches, to explore biomimetic strategies, such as cell membrane-derived vesicles [127], or the incorporation of peptide ligands to improve dual brain and neuron targeting. To the best of our knowledge, the use of inorganic materials in the microfluidic synthesis of DDSs for CNS disorders remains mostly uncovered. Collectively, these studies illustrate how microfluidics offers a transversal technological platform, applicable to multiple CNS pathologies, with the potential to accelerate translation from preclinical research toward clinical application.

In conclusion, the development of micro- and nanoparticles engineered through microfluidic technologies represents a highly promising therapeutic strategy for AD, owing to their capacity to simultaneously target multiple pathological hallmarks associated with the disorder: (i) inhibition of Aβ aggregation, and (ii) modulation of neuroinflammatory responses, providing neuroprotection. Furthermore, beyond their therapeutic benefits, the majority of the described nanosystems possess inherent diagnostic capabilities (magnetic, plasmonic, or fluorescent properties), which make them particularly well-suited for theranostic applications. For these reasons, microfluidically templated NPs represent innovative systems at the forefront of next-generation strategies for AD management.

### Routes of Drug Administration

Crossing the BBB remains the central challenge in treating neurodegenerative diseases. The BBB’s highly selective and restrictive permeability is essential for maintaining CNS homeostasis, as it tightly regulates the exchange of molecules between the bloodstream and neural tissue. However, this protective function also prevents most drugs from effectively reaching the brain through conventional routes, such as oral or parenteral administration [26,132]. To overcome this, innovative delivery systems are being explored, such as those based on nanotechnology [133]. These systems use various nanocarriers, including polymeric, lipid, and metal NPs, as well as liposomes and nanoemulsions, to enhance the solubility, stability, and brain uptake of drugs used in therapies. They can increase drug concentration in the brain, reduce dosage requirements and systemic side effects, and provide sustained drug release [101].

Among the various routes of drug administration, oral administration is the most common for chronic therapy due to its convenience and non-invasiveness; however, it is often limited in effectiveness due to several physiological and biochemical barriers [134]. First, the acidic stomach environment, the neutral to weakly alkaline conditions of the intestines, and digestive enzymes may degrade unstable compounds before they are absorbed. Poor aqueous solubility or low membrane permeability can also restrict uptake, resulting in inconsistent or low systemic exposure. In addition, drugs absorbed from the gastrointestinal tract must pass through the liver, where extensive hepatic metabolism can significantly reduce the amount that enters the systemic circulation (the ‘first-pass effect’). Finally, substantial variability in absorption and overall drug exposure is caused by large inter-individual differences driven by factors such as diet, gastric emptying and intestinal motility, co-medications, and genetic variability in metabolic enzymes [135,136]. Thus, these challenges are especially relevant in the context of CNS drugs, where only a small fraction of the absorbed compound may reach systemic circulation, and an even smaller portion can cross the BBB.

Among the other routes of drug administration, the parenteral ones, such as intravenous, subcutaneous, intramuscular, and transdermal administration, avoid gastrointestinal breakdown and first-pass metabolism, but introduce other significant limitations [137]. They are invasive, with injections potentially causing pain, local irritation, or infection if aseptic technique is not followed correctly, which may reduce patient adherence. Some parenteral techniques require trained personnel or clinic visits, which limit their feasibility for long-term or home therapy [138]. Moreover, the rapid clearance of many intravenously administered compounds often necessitates continuous infusion or repeated dosing, which can further burden the patient [139]. Transdermal delivery systems have also been developed to provide sustained systemic release of drugs through the skin. These patches offer the advantage of steady plasma levels, improved adherence, and non-invasiveness. However, their applicability is restricted to drugs with sufficient lipophilicity and low molecular weight, and skin irritation can occur with prolonged use.

One promising non-invasive approach that is now emerging is intranasal drug delivery. This method uses the anatomical pathways of the nasal cavity, primarily the olfactory and trigeminal nerves, to transport therapeutic agents directly to the brain [140], bypassing the systemic circulation and liver metabolism. The mechanism involves both intracellular and extracellular pathways. In the intracellular pathway, drugs are absorbed by olfactory sensory neurons and transported directly through axons. In the extracellular pathway, drugs diffuse through the nasal epithelium into the cerebrospinal fluid [141]. When drugs are encapsulated in NPs, they can greatly improve the delivery of drugs from the nose to the brain, enhance brain bioavailability and transport efficiency, offering a promising strategy for treating neurological disorders [142]. Despite these advances, significant challenges for industrial translation persist. These include high production costs, difficulties in scaling up manufacturing, and stability concerns. Collectively, these issues hinder clinical implementation. Indeed, alternative administration routes are also under investigation, including intrathecal and intracerebroventricular injections, which bypass the BBB entirely by delivering drugs directly into the cerebrospinal fluid. While these methods achieve high CNS drug concentrations, their invasiveness and associated risks restrict their use to severe or refractory cases.

Overall, the choice of administration route reflects a balance between drug properties, therapeutic goals, and patient compliance. Emerging delivery strategies, particularly those involving nanotechnology, hold promise to overcome the limitations of traditional approaches and expand the therapeutic arsenal for neurodegenerative diseases.

## 4. Conclusions and Outlooks

The BBB is a semipermeable and highly selective membrane, which is crucial in maintaining brain homeostasis by tightly controlling the entry of molecules into the CNS and preventing the passage of circulating toxins and pathogens into the brain [12]. Taking into consideration that nearly all large-molecule drugs do not penetrate the BBB and only 2% of small-molecule therapeutics can cross the BBB, much effort is still being made to devise novel brain-targeted delivery methods that penetrate the BBB efficiently to treat AD [27,143]. A promising solution to bridge this gap seems to be provided by micro- and nanomaterial technologies for the design and functionalization of NPs as advanced drug delivery tools for AD. These micro- and nanosystems leverage tailored designs to prevent, among others, insoluble protein aggregation and misfolded protein formation also in AD, thereby reducing aggregate-mediated neurotoxicity, improving drug delivery of different compounds, and limiting the potential side effects of treatments. Conventional bulk techniques for the synthesis of NPs warrant suitable nanocarriers that can incorporate, protect, and vehicle many different therapeutic compounds to the brain at relevant doses, by crossing the BBB and overcoming the limits related to the administration of free drugs, including the need for high frequency of pharmacological administration. However, some critical limitations remain when using conventional methodologies, such as poor reproducibility of the process, limited possibilities for material design, and scarce reproducibility.

A significant jump ahead in the tailored material synthesis of nanocarriers has been provided by the application of microfluidics to nanomedicine for future clinical translation and personalized therapies, also against AD. For instance, a tipping point in the conventional methods for NP synthesis is given by the introduction of advanced microfluidic platforms for accelerating and streamlining nanocarrier synthesis and overcoming the constraints and limitations of traditional bulk techniques, especially when industrial scale-up of NP synthesis to a GMP-conform process of production is required. Microfluidic systems thereby represent excellent tools for the generation of nanocarriers, providing unprecedented control over many parameters, including NP size, morphology, their balanced chemical composition, as well as the use of very low fluid volumes, with high levels of process reproducibility. For example, a novel Mixed Emulsion/Evaporation Technique (MEET) for nanogel synthesis developed by Limiti et al. [144], combined with microfluidic platforms [145], overcomes the reliance on hydrophilic/hydrophobic polymer systems that are required in traditional emulsification/evaporation processes. The resulting nanovector, as a consequence, shows remarkable stability and highly controlled drug release profiles. Overall, microfluidics would make brain-targeted drug delivery safer by improving the dosing difficulties and cheaper by reducing the financial burden for AD patients and their families, among others.

Yet, although their great potential, micro- and nanovectors present some limitations before their clinical translation, mainly related to the ideal route of administration to evade interactions with blood cells, the immune system, and complex dietary components, if ingested by oral administration, which may affect their stability and alter their advantageous features [120]. In the setting of AD, the final fate of micro- and nanovectors in the brain, once they release their cargo, is worthy of thorough attention, which depends mainly on their chemical composition. If not completely cleared in the neural milieu, their bioaccumulation is a crucial aspect that needs to be carefully evaluated to avoid neurotoxicity and neuroinflammatory responses that, in the long run, could contribute to and exacerbate the existing neurodegenerative condition.

In addition to the pressing necessity for more effective drug delivery strategies, there is an urgent need to develop better in vitro models of the BBB that can accurately simulate drug delivery in the CNS and disease pathology, with the aim of accelerating the development of novel therapeutics and understanding the pathogenesis of multifaceted neurological disorders, such as AD [143]. Traditional static 2D cell culture models fall short in providing complex and reliable in vitro platforms for addressing these needs. On the other hand, current animal models are indeed not appropriate in modeling all features of AD, and their usage is increasingly limited due to the ethical issues surrounding the use of animals in scientific research according to the 3R principles (Reduce, Refine, Replace) [146,147].

Taking into account that the BBB breakdown represents a triggering factor for AD pathogenesis, the advent of microfluidics to develop complex micro-engineered BBB systems (i.e., BBB-on-a-chip) for brain disease modeling and drug screening, as well as to streamline the design and production of nanocarriers may pave the way to novel opportunities for addressing issues about BBB-mediated pathophysiology of AD and for mimicking NP transport and distribution across the CNS [113,148,149]. Thus, thanks to these innovative microfluidic technologies, which can more accurately replicate the complex BBB structure and function to improve the predictive significance and the reproducibility of advanced in vitro models of AD, an alternative and effective research horizon is within reach. Some recent progress in human BBB-on-a-chip technologies is allowing scientists to gain new insights into the comprehension of the molecular mechanisms underlying BBB damage and dysfunction in AD pathogenesis, and how to exploit the BBB to treat AD [60,79,80,81,113,149]. To date, however, only a few BBB-on-a-chip platforms have been successfully applied to AD research, simultaneously integrating within the same device all the cellular players, molecular mechanisms, and hallmarks of the disease (such as neuroinflammatory and neurodegenerative pathways and cascades) by recapitulating the mechanistic aspects involved in the early onset and development of the pathology [31]; therefore, much work needs to be done. Besides this issue, as with any other OoC model, a crucial point will be the standardization of regulatory testing, methods and procedures for the reproducibility of suitable BBB-on-a-chip models. Indeed, establishing validation and qualification criteria for developing these preclinical platforms as alternative research approaches, in terms of reproducibility and predictive capacity, based on the purpose of the study, would be a future matter of debate according to the directives of the EU regulatory policy makers. Although these in vitro platforms offer clear advantages over in vivo animal models, such as greater robustness at reduced costs, the option to use human cells for more accurate replication of pathophysiological traits, and enhanced reproducibility and throughput with no ethical issues, they are still unable to capture the full complexity of a living organism as a whole, thereby laying the groundwork for further research in the near future [36].

Altogether, these microfluidic-based strategies offer fast-paced technological advances in biomedical research to efficiently fabricate micro- and nanocarriers for targeted drug delivery into the CNS together with reliable human BBB-on-a-chip devices for mimicking neurodegenerative conditions, and investigating receptor mediated uptake mechanisms of micro- and nanovectors, as well as a compelling path forward for crossing the BBB and developing more effective and targeted therapies for neurodegenerative diseases, including AD. As a matter of fact, these nanomedicine and microfluidic-based approaches may have a dual-purpose use and represent a significant stride towards the provision of newer customized nanocarriers to cross the BBB for clinical applications, as well as alternative microphysiological BBB models to an industrial level for several key applications. Such advances will mark the spot for the search and development of new technological and tissue engineering strategies that may help tackle AD and reduce the burden of the disease worldwide.

## Figures and Tables

**Figure 1 ijms-26-09478-f001:**
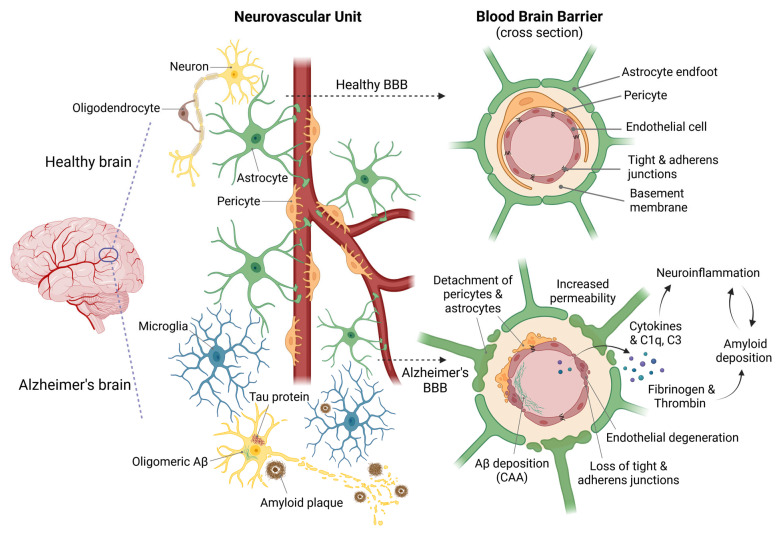
Alterations of BBB integrity in AD. In a healthy brain (top), the NVU maintains the integrity of the BBB. This promotes a healthy brain microenvironment characterized by physiological function, cell-to-cell communication, adequate blood flow and nutrient supply, and low levels of inflammation. By contrast, pathological features characterize the AD’s brain (bottom), such as the accumulation of NFTs of hyperphosphorylated tau protein and the accumulation of intracellular Aβ oligomers, and the deposition of extracellular Aβ plaques. These events are accompanied by activated microglia, resulting in the release of pro-inflammatory mediators. On the right, a detailed cross-sectional view of the BBB illustrates the NVU’s structural organization, emphasizing the presence of tight and adherens junctions between vascular endothelial cells. These junctions are critical for maintaining barrier integrity and selective permeability. In AD, the pathological condition is characterized by altered intercellular communication between endothelial cells in blood vessels, pericytes, and astrocytes. This results in the degeneration of the NVU and significantly disrupts the BBB. Increased BBB permeability allows plasma components such as fibrinogen, thrombin, complement proteins (C1q and C3), and cytokines to infiltrate brain tissue. These molecules promote further Aβ deposition and exacerbate neuroinflammation, thus perpetuating BBB damage and neurodegeneration. [Created with BioRender.com].

**Figure 2 ijms-26-09478-f002:**
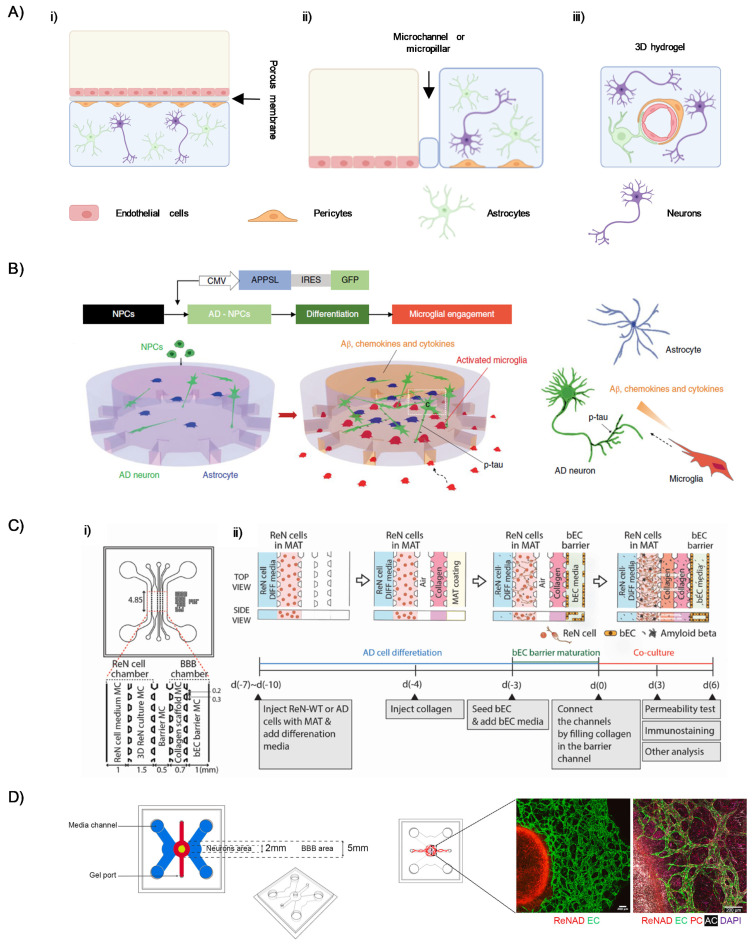
(**A**) Schematic representation of three microfluidic device configurations: (**i**) vertical, (**ii**) parallel, and (**iii**) full-contact. (**B**) Schematics showing multicellular 3D layouts in a microfluidic human AD culture model (reprinted with permission from [78]). (**C**) (**i**) Schematic of a microfluidic parallel configuration highlighting the ReN-AD and BBB chambers, separated by an intervening barrier microchannel (MC), and (**ii**) experimental protocol for AD model development (reprinted with permission from [79]). (**D**) (**Left**): Schematic of a microfluidic platform used to create a BBB in a circular compartment (red) surrounding a central neural compartment (yellow), allowing direct communication. Side media channels (blue) are used to perfuse the BBB by creating a pressure gradient across the central compartments. (**Right**): Confocal projected z-stack of a portion of the full device with immunofluorescence staining after 30 days of co-culture (reprinted with permission from [80]).

**Table 2 ijms-26-09478-t002:** Summary of representative microfluidic BBB-on-a-chip models used to study AD. The number of dots indicates the qualitative complexity of the structure of the devices, with more dots indicating a more complex structure.

Device Configuration	Application	Complexity		REF
		Technological	Biological	
Parallel (Cylindrical)	Microglia recruitment, neuroinflammation, Aβ accumulation, and p-tau aggregation	•	•	[78]
Parallel	BBB disfunction and Aβ deposition	••	••	[79]
Full-contact	High-resolution tracking of Alzheimer’s-related changes in BBB function	•••	•••	[80]
Full-contact	Pathogenic mechanism of APOE4 expression in pericytes	•	•••	[81]
Full-contact	Diabetes mellitus contribution to Alzheimer’s disease	••	••	[82]
Vertical design	Immune-brain interaction in Alzheimer’s disease	•	••	[83]

**Table 3 ijms-26-09478-t003:** Summary of nano- and microparticle drug delivery systems for CNS disorders (with particular focus on AD therapy) synthesized via microfluidic approaches.

		Material Class	Composition	Target	Mechanism of Action	Biological Validation		REF
						In Vitro	In Vivo	
**Alzheimer’s disease**	Nanoparticles	Inorganic NPs	Chitosan coated Gold nanorods & Spion NPs functionalized with peptide D1	Reduced Aβ deposition and plaque formation	Binding of Aβ through Ab-D1 interaction	SH-SY5Y neuroblastoma cells in a 2D model	-	[116]
		Organic NPs	Lipid nanoparticles co-encapsulating a-mangostin (a-M) and b-site APP cleaving enzyme 1 (BACE1) siRNA	Neuroprotection & gene silencing to reduce Aβ synthesis	Delivery of a-M and BACE1 SiRNAs	BV-2, PC-12, and Calu-3 cell lines in 2D and transwell model	Male APP/PS1 mice and Male C57BL/6 mice	[117]
		Hybrid NPs	Chitosan & graphene quantum dots	Neuroprotection and neuroinflammation	anti-inflammatory effect and regulation of the brain tissue microenvironment	C6 glioma cells (2D model) & bEnd.3, astrocytes and BV-2 glial cells in a transwell model	Sprague-Dawley Male rats	[118]
		Organic NPs	Pantothenate encapsulated in transferrin modified liposomes (Pan@TRF@Liposome NPs)	Neuroinflammation, metabolic disfunction, neuronal death	modulation of CRM1-mediated PKM2 nuclear translocation	BV-2 glial cells in a 2D model	Male APP/PS1 mice and C57BL/6J mice	[119]
	Microparticles	Hydrogel microparticles	alginate-dopamine core-shell microcapsules	Reduced Aβ deposition, and acetylcholinesterase (AChE) activity & neuroprotection	Delivery of Donepezil	N2A neuroblastoma cells and 3T3 cells in 2D models	APP/PS1 mice and C57BL/6 mice	[120]
		Hydrogel microparticles	PLGA-GelMa core-shell microcapsules	Reduced Aβ deposition & neuroprotection	Delivery of Ginkgo biloba extract	N2A neuroblastoma cells and 3T3 cells in 2D models	APP/PS1 mice and C57BL/6J mice	[121]
		Hydrogel microparticles & inorganic NPs	Mesoporous silica nanocarriers encapsulated into alginate-dopamine core-shell microcapsules	Reduced Aβ deposition and AChE activity	Delivery of Galantamine hydrobromide	N2A, Caco-2 and 3T3 cells in a 2D model	APP/PS1 mice and C57BL/6 mice	[122]
		**Material class**	**Composition**			**Application**		**REF**
**Other CNS disorders**		Organic NPs	PEGylated lipid nanoparticles loaded with antisense oligonucleotides			Neuronal gene silencing		[123]
		Hybrid NPs	PLGA core with lamotrigine, lipid shell functionalized with D-T7 and Tet1 peptides			Epilepsy therapy via dual BBB and neuron targeting		[124]
		Hybrid NPs	PLGA core with lamotrigine, lipid shell modified with peptides (T7, D-T7, GSH, TGN, CGN, TAT)			BBB penetration and epilepsy therapy		[125]
		Organic NPs	Docosahexaenoic acid liposomes			Glioblastoma therapy		[126]
		Organic NPs	Macrophage cell membrane-derived nanovesicles loaded with indocyanine green			Targeted brain tumor theranostics		[127]
		Organic NPs	Solid lipid nanoparticles loaded with brain-derived neurotrophic factor			Brain delivery for post-traumatic brain injury neuroinflammation		[128]
		Organic NPs	Lipid nanoparticles delivering mRNA encoding IL-10			Stroke therapy, BBB protection, neuroinflammation modulation		[129]
		Organic NPs	Multivesicular liposomes containing gastrodin			Oral delivery across BBB for CNS diseases		[130]
		Organic NPs	Polymeric nanoparticles (cross-linked BSA) conjugated with RVG peptide			Targeted brain delivery for seizure control and CNS disorders		[131]

## Data Availability

No new data were created or analyzed in this study. Data sharing is not applicable to this article.

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
