# Peer review of "Microfluidic-Based Technologies for Crossing the Blood–Brain Barrier Against Alzheimer’s Disease: Novel Strategies and Challenges"

_ijms, 2025, doi:10.3390/ijms26199478_

Round 1
Reviewer 1 Report
Comments and Suggestions for Authors
The paper presents a comprehensive review of blood–brain barrier models, particularly microfluidic BBB-on-a-chip systems, in the context of Alzheimer’s disease (AD) pathophysiology and therapeutic development. It touches on a wide range of literature, beginning with the neurovascular contributions to AD, the limitations of existing in vivo and in vitro models, and the emergence of organ-on-a-chip (OoC) and microfluidic platforms as more physiologically relevant tools. Overall, the writing is clear and well-organized, with logical transitions between topics. Figures are well-conceived.
There are few point of improvement for the manuscript:
- Discussion of regulatory and ethical pathways
For emerging technologies like BBB-on-a-chip to have clinical impact, validation against regulatory standards (FDA, EMA) is crucial. The review could benefit from a section on ongoing regulatory qualification efforts for OoC platforms.
2. Mechanistic insight into model limitations
The review acknowledges that organoids lack perfusion and shear stress but does not delve into quantitative data on how these deficits alter BBB function (e.g., reduced expression of specific transporters or enzymes). More mechanistic evidence would strengthen the argument.
Author Response
We would like to thank the Reviewer for taking the time to review this manuscript and provide such valuable feedback that will help us to significantly improve the quality of our work. We hope this revised manuscript satisfactorily addresses the Reviewer’ comments. Below is a point-to-point reply to all the comments raised by the Reviewer; in the main text all the relevant changes are now highlighted to facilitate the revision.
Reviewer #1:
The paper presents a comprehensive review of blood–brain barrier models, particularly microfluidic BBB-on-a-chip systems, in the context of Alzheimer’s disease (AD) pathophysiology and therapeutic development. It touches on a wide range of literature, beginning with the neurovascular contributions to AD, the limitations of existing in vivo and in vitro models, and the emergence of organ-on-a-chip (OoC) and microfluidic platforms as more physiologically relevant tools. Overall, the writing is clear and well-organized, with logical transitions between topics. Figures are well-conceived. There are few point of improvement for the manuscript:
Comment 1: Discussion of regulatory and ethical pathways.
For emerging technologies like BBB-on-a-chip to have clinical impact, validation against regulatory standards (FDA, EMA) is crucial. The review could benefit from a section on ongoing regulatory qualification efforts for OoC platforms.
Response 1: We thank the Reviewer for pointing this out. Therefore, we have added a new paragraph 2.3 (highlighted in red) in the manuscript to address the regulatory and ethical issues of OoC.
Comment 2: Mechanistic insight into model limitations
The review acknowledges that organoids lack perfusion and shear stress but does not delve into quantitative data on how these deficits alter BBB function (e.g., reduced expression of specific transporters or enzymes). More mechanistic evidence would strengthen the argument.
Response 2: We thank the Reviewer for this comment, and we agree with it. We have now, accordingly, revised Section 2.1 to improve clarity and expand the discussion on the role of shear stress in the structural and functional development of the BBB. In particular, we have modified the text relating to Transwell systems and organoid models to emphasise the importance of flow conditions. The text now more clearly describes how shear stress contributes to the maturation of endothelial cells and the formation of tight junctions, both of which are critical for establishing a physiologically relevant BBB model.
Reviewer 2 Report
Comments and Suggestions for Authors
The authors in this manuscript have conducted a thorough review of microfluidic technologies in crossing the BBB in alzheimer's disease. The manuscript is very well written however, the following can be improved in the manuscript to make it a more comprehensive review article
1. Drug delivery systems are summarized in this manuscript; it needs to be expanded to include studies performed for each type and the results of the studies, as the table is limited and contains only a few examples.
2. Routes of administration also need to be further explored in the literature.
3. In vitro physicochemical characterization of microfluidic systems needs further explanation and expansion in the manuscript to make it fully comprehensive
Author Response
We sincerely appreciate the Reviewer for dedicating their time to evaluate this manuscript and for providing insightful feedback, which has been instrumental in enhancing the quality of our work. We believe that the revised manuscript effectively addresses the Reviewer’ comments. Below, we provide a detailed, point-by-point response to each comment. All corresponding changes in the main text have been highlighted to facilitate the review process.
Reviewer 2
The authors in this manuscript have conducted a thorough review of microfluidic technologies in crossing the BBB in Alzheimer's disease. The manuscript is very well written however, the following can be improved in the manuscript to make it a more comprehensive review article.
Comment 1: Drug delivery systems are summarized in this manuscript; it needs to be expanded to include studies performed for each type and the results of the studies, as the table is limited and contains only a few examples.
Response 1: We thank the reviewer for the insightful comment.
Table 3 has been revised and now includes 9 recent studies (from 2022 to 2025) which explore the use of microfluidically synthesized nano-DDSs for the treatment of several CNS disorders. This update allowed us to expand the discussion on microfluidics as cutting-edge technology for the synthesis of nano-DDSs, while emphasizing the BBB-crossing ability as a key feature of the analyzed systems.
Moreover, a brief discussion of main findings from these studies has been included in the new version of the manuscript.
Comment 2: Routes of administration also need to be further explored in the literature.
Response 2: We thank the reviewer for bringing this to our attention. In response, we have added Section 3.1, which specifically discusses the various routes of drug administration and their limitations.
Comment 3: In vitro physicochemical characterization of microfluidic systems needs further explanation and expansion in the manuscript to make it fully comprehensive.
Response 3: We agreed with the reviewer comment. In fact, as we highlighted in the manuscript, microfluidics is emerging as a crucial technology for the fine control e definition of NPs physicochemical characteristics. In line with the reviewer’s suggestion, we have revised the manuscript by adding information on size, PDI, and zeta potential for the discussed studies.
Round 2
Reviewer 2 Report
Comments and Suggestions for Authors
The authors have responded to all my suggestions, the paper is now fit for publication